# Quality Assessment of Endoscopic Forceps Biopsy Samples under Magnifying Narrow Band Imaging for Histological Diagnosis of Cervical Intraepithelial Neoplasia: A Feasibility Study

**DOI:** 10.3390/diagnostics11020360

**Published:** 2021-02-20

**Authors:** Kunihisa Uchita, Hideki Kobara, Kenji Yorita, Yuriko Shigehisa, Chihiro Kuroiwa, Noriko Nishiyama, Yohei Takahashi, Yuka Kai, Jun Kunikata, Toshio Shimokawa, Uiko Hanaoka, Kenji Kanenishi, Tsutomu Masaki, Koki Hirano, Noriya Uedo

**Affiliations:** 1Department of Gastroenterology, Kochi Red Cross Hospital, Kochi 780-8562, Japan; ucchy31@yahoo.co.jp (K.U.); yurikonoguchi630@gmail.com (Y.S.); kuroiwa.kochi@gmail.com (C.K.); 2Department of Gastroenterology and Neurology, Faculty of Medicine, Kagawa University, Kagawa 761-0793, Japan; n-nori@med.kagawa-u.ac.jp (N.N.); tmasaki@med.kagawa-u.ac.jp (T.M.); 3Department of Diagnostic Pathology, Kochi Red Cross Hospital, Kochi 780-8562, Japan; kenjiyorita@gmail.com; 4Department of Gynecology, Kochi Red Cross Hospital, Kochi 780-8562, Japan; yoheitaka@gmail.com (Y.T.); memento-mori@mbr.nifty.com (Y.K.); hirarin0908@k7.dion.ne.jp (K.H.); 5Department of Clinical Research Support Center, Faculty of Medicine, Kagawa University, Kagawa 761-0793, Japan; kunikata@med.kagawa-u.ac.jp; 6Department of Clinical Study Support Center, Wakayama Medical University, Wakayama 641-8510, Japan; toshibow2000@gmail.com; 7Department of Gynecology, Faculty of Medicine, Kagawa University, Kagawa 761-0793, Japan; uhanaoka@med.kagawa-u.ac.jp (U.H.); kane@med.kagawa-u.ac.jp (K.K.); 8Department of Gastrointestinal Oncology, Osaka International Cancer Institute, Osaka 541-8567, Japan; noriya.uedo@gmail.com

**Keywords:** uterine cervical neoplasms, cervical intraepithelial neoplasia, biopsy, endoscopy, colposcopy

## Abstract

The current standard for diagnosing cervical intraepithelial neoplasia (CIN) is colposcopy followed by punch biopsy. We have developed flexible magnifying endoscopy with narrow band imaging (ME-NBI) for the diagnosis of CIN. Here, we investigated the feasibility of targeted endoscopic forceps biopsy (E-Bx) under guidance of ME-NBI for the diagnosis of CIN. We prospectively enrolled 32 consecutive patients with confirmed or suspected high-grade CIN undergoing cervical conization. Next to colposcopy, the same patients underwent ME-NBI just before conization. ME-NBI was performed, and 30 E-Bx samples were taken from lesions suspicious for high-grade CIN and 15 from non-suspicious mucosa. We recalled 82 punch biopsy (P-Bx) specimens taken from lesions suspicious for high-grade CIN under colposcopic examination before enrollment. The proportion of sufficient biopsy samples, which had an entire mucosal layer with subepithelial tissue, for the diagnosis of CIN was evaluated by both methods. Performance of targeted E-Bx for the final diagnosis of at least high-grade CIN was calculated. Seventeen P-Bx specimens were unavailable. The proportion of sufficient samples with E-Bx was 84%, which was similar to that with P-Bx (87%) (*p* = 0.672). The sensitivity, specificity, and accuracy of ME-NBI using E-Bx was 92%, 81%, and 88%, respectively. In conclusion, ME-NBI-guided E-Bx samples were feasible for histological diagnoses of CIN, and further investigation of its diagnostic accuracy is warranted.

## 1. Introduction

Colposcopy followed by punch biopsy (P-Bx) is usually performed for further examination of positive Pap smear tests, such as for low-grade or high-grade squamous cell intraepithelial lesions [1]. However, diagnostic accuracy of colposcopy for diagnosing cervical intraepithelial neoplasia (CIN) is often insufficient [2]. Narrow band imaging (NBI) is an optical–digital technique used for the diagnosis of superficial neoplasms in the digestive tract. This technique contrasts vasculature and surface structure of the superficial mucosa in the digestive tract and improves the diagnostic yield of endoscopy for the identification and characterization of superficial neoplasms [3,4]. Moreover, in combination with magnifying endoscopy (ME), it visualizes microvascular architecture and microsurface structure that correspond with histology. We have developed a novel method that uses flexible gastrointestinal ME-NBI for the diagnosis of CIN [5]. We found that the ME-NBI showed the characteristic findings of CIN better than colposcopy [6].

Under colposcopic observation, P-Bxs are carried out for histological diagnoses of CIN. However, the procedure sometimes fails to acquire samples from appropriate sites because of the long distance between the colposcopy lens and the lesion, and the procedure requires skillful maneuvering of the P-Bx forceps and vaginal speculum [7]. Further technical developments are expected to overcome these issues for tissue sampling [8,9]. In ME-NBI diagnosis, an examiner can freely move the endoscope close to the lesion. Moreover, biopsy forceps are inserted through the working channel which is situated next to the endoscopy lens, enabling precise targeted biopsy of the lesion under direct vision. In the present study, we evaluated whether endoscopic forceps biopsy (E-Bx) samples under ME-NBI were feasible for the histological diagnosis of CIN.

## 2. Materials and Methods

### 2.1. Study Design, Setting and Participants

This observational study was conducted between January 2017 and September 2019 at two medical centers in Japan: Kochi Red Cross Hospital and Kagawa University Hospital. The study protocol was approved by the Institutional Review Board in each institution. The study was registered in the University Hospital Medical Information Network Clinical Trials Registry as Number 000021142. We prospectively enrolled 32 consecutive patients who had been scheduled to undergo conization for confirmed or suspected high-grade CIN (≥CIN2), based on colposcopic findings or histological findings of P-Bx. Along with colposcopy, the same patients underwent ME-NBI just before conization. All of these patients gave written informed consent for study participation. The examination flowchart and diagram of study enrollment are shown in Figure 1a,b, respectively.

The Clinical Ethics Committee of Kochi Red Cross Hospital (Approval No.198, 17 February 2016) and Kagawa University Hospital (Approval No.H28-043, 21 July 2016) approved this study.

### 2.2. Colposcopic Procedure

Before study enrollment, the patients underwent colposcopic examination at the gynecology department in the same hospital or referral hospital. Following the conventional method, each patient was placed in the lithotomy position and the cervix was observed with colposcopy (Figure 2a), and P-Bx samples were taken from ≥CIN2 lesions using punch biopsy forceps (Figure 2b). The diagnostic criteria for ≥CIN2 were dense acetowhite epithelium, coarse mosaic, and coarse punctation, based on the Rio 2011 Colposcopy Nomenclature of the International Federation of Cervical Pathology and Colposcopy [10]. Random biopsies were not performed in this study.

### 2.3. Endoscopic Procedure

Endoscopy was performed by three endoscopists (K.U., N.N. and H.K.) who had experience of >50 cases of endoscopic diagnoses of the uterine cervix. A magnifying videoendoscope (EVIS GIF H260Z or H290Z; Olympus Medical Systems, Tokyo, Japan), a video processor (EVIS CV-290; Olympus Medical Systems) and a light source (EVIS CV-290; Olympus Medical Systems) that work in the NBI mode were used for all procedures. A balloon occlusion device (Fuji Systems, Tokyo, Japan) was attached to the videoendoscope to avoid water leakage during water immersion observation. The cervix was cleaned by water jet from the endoscope (Figure 3a). In the NBI mode, the whole cervix was observed and suspected ≥CIN2 lesions were identified. Colposcopic findings obtained by previous colposcopic examination were masked in endoscopists who examined the cervix. In areas suspicious for ≥CIN2, magnifying observation was performed as necessary (Figure 3b). The diagnostic criteria for the ≥CIN2 lesions were based on our previous studies as follows: the presence of thick white epithelium or thin white epithelium plus atypical vessels or dense acetowhite epithelium (W2) [5,6]. White epithelium was defined as thin white epithelium when the underlying vessels were visible, and thick white epithelium when the underlying vessels were invisible. Atypical vessels were defined as microvessels that satisfied more than two of the following four conditions: dilatation, crawling, irregular arrangement, and caliber change according to the microvascular classification of early esophageal neoplasms with squamous epithelium [11]. After endoscopic diagnosis, disposable endoscopic biopsy forceps (Radial Jaw 4; Boston Scientific Japan, Tokyo, Japan) (Figure 4a) were inserted through a working channel of the videoendoscope, and E-Bx samples were taken from the ≥CIN2 lesions under direct vision (Figure 4b). Biopsy specimens were also taken from non-neoplastic cervical mucosa.

### 2.4. Evaluation of Biopsy Specimens

Each biopsy sample was immersed in 10% formalin, embedded in paraffin, and sliced for histological examination. Diagnosis of CIN was based on WHO classification [12] by a single experienced pathologist (Y.K.). The pathologist was informed about whether the tissue sample was E-Bx or P-Bx.

### 2.5. Outcome Measures

For accurate histological diagnosis of CIN, it is better to include the entire epithelial layer and part of the subepithelial interstitium in the biopsy specimen [8]. Accordingly, as a primary outcome, the proportion of sufficient samples was compared between E-Bx and P-Bx. When both the entire mucosal layer and subepithelial interstitium were present in the biopsy sample, the sample was defined as sufficient (Figure 5a), otherwise it was defined as insufficient (Figure 5b,c). Secondary outcomes were: differences in maximum diameter of biopsy samples between the E-Bx and P-Bx groups; diagnostic performance (sensitivity, specificity, positive predictive value (PPV) and negative predictive value (NPV)); accuracy of ME-NBI using E-Bx for diagnosing ≥CIN2 lesions; and difference in PPV between ME-NBI using E-Bx and colposcopy using P-Bx. 

### 2.6. Statistical Analysis

Continuous data are presented as mean ± standard deviation (SD). The rates of all outcomes were calculated with 95% confidence intervals (CIs). A two-sided Fisher’s exact test was used to compare the proportion of evaluable samples of E-Bx and P-Bx. The specimen size was compared between the groups using a paired *t*-test. The PPV of ME-NBI using E-Bx vs. colposcopy using P-Bx for CIN2 was compared using a chi-squared test. A *p*-value <0.05 was considered statistically significant. All statistical analyses were conducted using JMP version 9.0 (SAS Institute Inc., Cary, NC, USA).

## 3. Results

### 3.1. Participants and Descriptive Data

A total of 32 patients underwent ME-NBI examination after colposcopy and before conization, and 45 E-Bx tissue samples were obtained, consisting of 30 from lesions suspicious for high-grade CIN (≥CIN2) and 15 from non-suspicious mucosa based on the ME-NBI findings (Figure 1). ME-NBI did not identify ≥CIN2 lesions in two of 32 patients; therefore, there were 30 E-Bx samples from lesions suspicious for ≥CIN2. Increased biopsy number may potentially increase cervical bleeding, and poor visualization of the cervix due to the bleeding may influence the following conization. Therefore, E-Bx samples for non-suspicious mucosa resulted in 15 samples. Eighty-two P-Bx samples that were taken from ≥CIN2 lesions before study enrollment were collected from the Kochi Red Cross Hospital, Kagawa University Hospital, and referral hospitals. Histological slides were unavailable in 17 samples; therefore, the remaining 65 samples were included in the analysis (Figure 1b). The descriptive data of the study participants are shown in Table 1. The mean age was 43 ± 10 years. Final histology of the conization specimens were CIN3 in 24 patients, CIN2 in three, CIN1 in four, and no neoplasm in one.

### 3.2. Outcome Data

The proportion of sufficient samples showed no significant difference between E-Bx and P-Bx groups (*p* = 0.672) (Table 2). The sufficient sample with both the entire epithelial layer and part of the subepithelial interstitium in E-Bx is represented in Figure 5a. The mean (±SD) maximum diameter of E-Bx samples (1.7 ± 0.81 mm) was significantly smaller than that of P-Bx samples (5.1 ± 2.2 mm; *p* < 0.001) (Table 2). Among the insufficient samples, all eight P-Bx samples contained at least the whole epithelial layer, whereas three of seven (43%) E-Bx samples did not contain the whole epithelial layer (Table 3). There was a lack of subepithelial interstitium (Figure 5b) in four of seven (57.1%) E-Bx and in all eight P-Bx samples. The entire mucosal layer and subepithelial interstitium were both absent in three of seven (42.9%) E-Bx samples (Figure 5c) and none of the eight P-Bx samples, suggesting a significantly lower rate of samples without both the entire mucosal layer and subepithelial interstitium in P-Bx than in E-Bx (*p* = 0.038). The cut-off value of specimen size needed for successful sampling was 1.7 mm (Figure 6). The diagnostic performance of ME-NBI using E-Bx for ≥CIN2 is summarized in Table 4. The PPV of ME-NBI using E-Bx (89%) was significantly higher than that of colposcopy using P-Bx (51%, *p* < 0.001).

## 4. Discussion

This is believed to be the first study to evaluate the quality of E-Bx samples under ME-NBI in the histological diagnosis of CIN. Although the specimen size of the E-Bx samples was significantly smaller than that of P-Bx samples, the sufficiency of biopsy samples to diagnose CIN was similar, and E-Bx under ME-NBI showed promising diagnostic performance for ≥CIN2.

We found that, despite their small specimen size, most E-Bx samples contained both the entire epithelial layer and subepithelial tissue; thus, the diagnosis of CIN was equally possible with E-Bx and P-Bx. Therefore, E-Bx under ME-NBI could be an acceptable method for diagnosing cervical cancer. P-Bx uses forceps with large mechanical claws, which often causes painful discomfort to patients and has the potential risk of delayed bleeding. A previous study showed that mean maximum diameter obtained from subepithelial tumors in the digestive tract using E-Bx forceps was 1.8 mm [13]. This is similar to the cut-off value of 1.7 mm for the size of evaluable specimen in the present study. Although larger biopsy samples are reported to have better diagnostic accuracy than small samples [14], the diagnostic goal of colposcopy is the precise identification of CIN, and targeted P-Bx is necessary for accurate histological diagnosis [15]. Accordingly, larger specimens obtained by invasive P-Bx may not be mandatory for CIN diagnosis, thus less invasive techniques and devices must be developed [16]. However, we found that some samples in the E-Bx group could not be evaluated histologically because of the small size of the forceps cup (2.2 mm). The small size of the forceps meant that they tended to slip on the surface because of the hardness and roundness of the cervical mucosa. Once the mucosa is fixed by the central claw of the forceps, it can be captured without slipping by closing the opened forceps slowly. Therefore, a suitable size of biopsy forceps without the above technical issues is expected to be developed. 

For comparison of diagnostic performance between ME-NBI using E-Bx and colposcopy using P-Bx, PPV was compared because P-Bx samples were only taken from ≥CIN2 lesions. The PPV for ME-NBI (89%) was significantly higher than that for colposcopy (51%, *p* < 0.001). One of the reasons why PPV for ME-NBI was better was because ME-NBI using E-Bx could be performed under direct endoscopic vision. The E-Bx forceps emerged from a working channel of endoscope next to the objective lens; therefore, suspicious areas could be more precisely and easily targeted than with P-Bx under colposcopy. In the field of gastrointestinal endoscopy, targeted biopsy under endoscopic direct vision has been standardized since the 1970s [17], demonstrating excellent outcomes with the combination of image-enhanced technology [18,19]. Image quality and the accuracy of endoscopic biopsies have led to high diagnostic performance of ME-NBI despite small sizes of obtained tissue samples. According to a meta-analysis, the sensitivity of colposcopy was good at 91%, but the specificity was only 24.6% [7]. Compared with the data in the meta-analysis, ME-NBI in the present study showed higher specificity of 81.3%. P-Bx under colposcopy is usually performed for lesions with acetowhite epithelium. If there are no lesions that are suspicious for CIN, multiple random biopsies are often attempted in order to avoid overlooking CIN [20,21,22]. This strategy probably explains the reason why colposcopy shows high sensitivity but low specificity. ME-NBI improves the diagnostic accuracy of biopsy and may reduce the number of biopsies from areas with uncertain endoscopic findings. ME-NBI using E-Bx has probably become helpful for diagnoses of CIN. 

The instrumental cost of E-Bx under ME-NBI and P-Bx under colposcopy is estimated to be almost equal. E-Bx under ME-NBI which may show high diagnostic performance and has a potential benefit to reduce the number of biopsy times, leading to its cost-effectiveness over colposcopy-guided P-Bx. Furthermore, patients’ pain accompanied by biopsy may be lower in E-Bx under ME-NBI. Disposable E-Bx forceps would also be advantageous in the view of infection control. A further study is ongoing to clarify these potentials.

This study had some limitations. Firstly, the participants were limited to patients undergoing cervical conization; CIN1 and non-cancerous lesions were excluded. Secondly, only the diagnostic performance of ME-NBI using E-Bx for CIN was evaluated. P-Bx samples under colposcopy were only acquired from lesions suspicious for ≥CIN2, therefore PPV alone was statistically calculated for the diagnostic ability of colposcopy. Thus, the detailed comparison of ME-NBI vs. colposcopy was impossible. A prospective randomized controlled study of ME-NBI using E-Bx vs. colposcopy using P-Bx should be conducted to establish the true ability of ME-NBI. Thirdly, only a small number of biopsy samples was included. 

In conclusion, this study has demonstrated that E-Bx samples under ME-NBI were feasible for the histological diagnosis of CIN. The results warrant further investigation of the diagnostic yield of ME-NBI for CIN in patients with abnormal Pap smears.

## Figures and Tables

**Figure 1 diagnostics-11-00360-f001:**
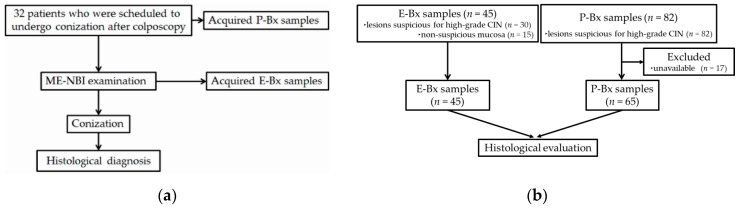
(**a**) Examination flowchart; (**b**) diagram of study enrollment. ME-NBI, magnifying endoscopy with narrow band imaging; P-Bx, punch biopsy; E-Bx, endoscopic forceps biopsy; CIN, cervical intraepithelial neoplasia.

**Figure 2 diagnostics-11-00360-f002:**
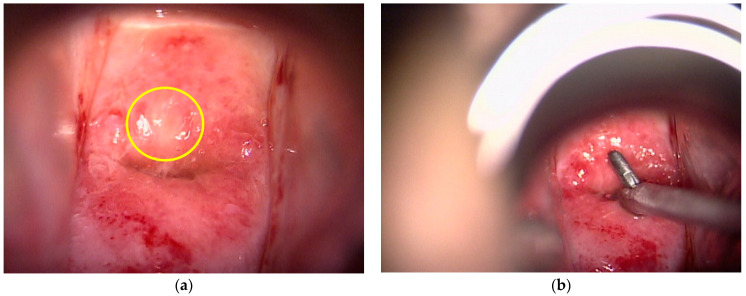
Colposcopic finding of cervix. (**a**) Colposcopic finding of cervical intraepithelial neoplasia showing the thin acetowhite epithelium (yellow circle); (**b**) punch biopsy with colposcopy. Field of vision was partially blocked by biopsy forceps.

**Figure 3 diagnostics-11-00360-f003:**
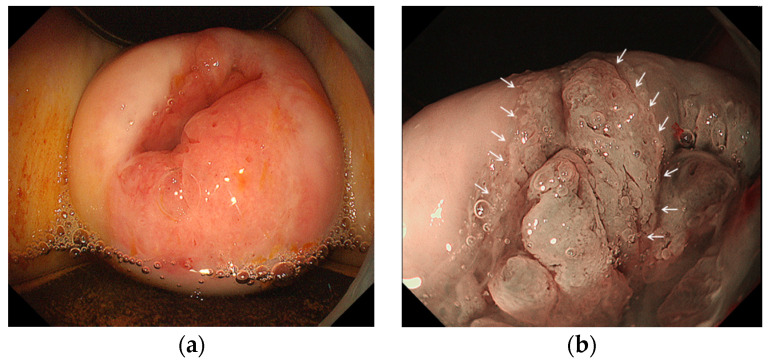
Endoscopic finding of cervix. (**a**) Conventional endoscopic finding with white light imaging. Cervical intraepithelial neoplasia (CIN) could not be detected; (**b**) Magnifying endoscopic finding with narrow band imaging showing a thick white epithelium of CIN3 (white arrows).

**Figure 4 diagnostics-11-00360-f004:**
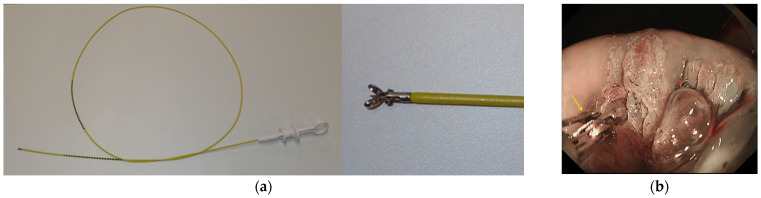
(**a**) Endoscopic biopsy forceps (Radial Jaw 4; Boston Scientific, Tokyo, Japan). The cup size of the forceps is 2.2 mm. (**b**) Targeted biopsy under endoscopic direct vision. Yellow arrow shows endoscopic biopsy forceps.

**Figure 5 diagnostics-11-00360-f005:**
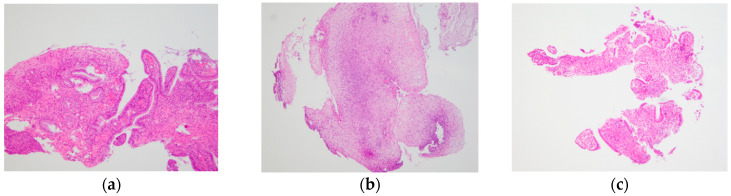
Each picture shows the histological finding of either a sufficient sample or insufficient sample. (**a**) Histological finding of a sufficient sample, which had the entire mucosal layer with subepithelial interstitium; (**b**) histological finding of an insufficient sample, which had only the entire mucosal layer; (**c**) histological finding of an insufficient sample, which lacked the entire epithelial layer and subepithelial interstitium, and therefore an accurate diagnosis could not be made.

**Figure 6 diagnostics-11-00360-f006:**
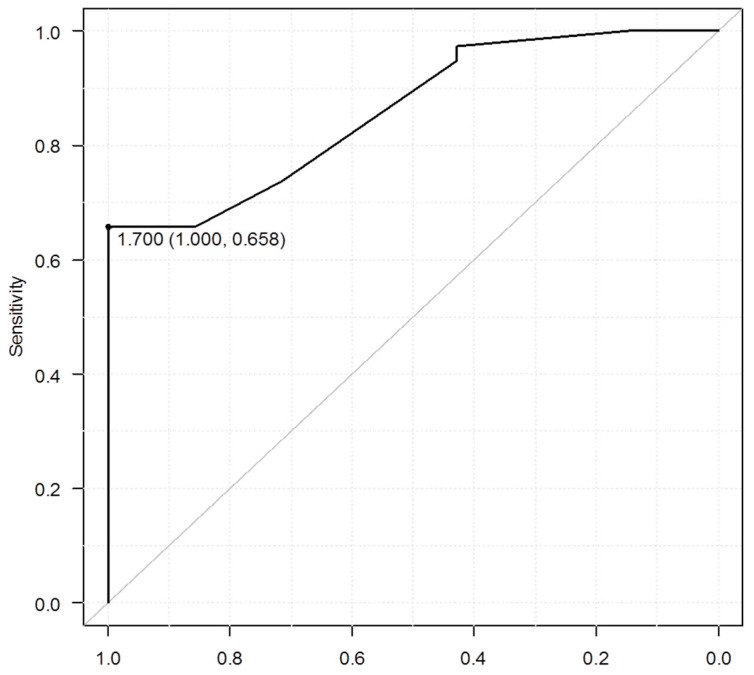
Optimal cutoff value of successful sample by receiver operating characteristic curve and Youden index, cut-off value was 1.7 mm.

**Table 1 diagnostics-11-00360-t001:** Clinicopathological data of patients who underwent cervical conization.

Total no. of patient	32
Mean age (SD)	43 (10) years
No. of P-Bx	82
Positive (≥CIN2)	42
Negative (≤CIN1)	40
Final pathological diagnosis of surgical specimens	
≥CIN3	24
CIN2	3
CIN1	4
No malignancy	1

CIN, cervical intraepithelial neoplasia; P-Bx, punch biopsy; SD, standard deviation.

**Table 2 diagnostics-11-00360-t002:** Results of outcomes regarding sample quality for E-Bx and P-Bx.

	E-Bx (*n* = 45)	P-Bx (*n* = 65)	*p*-Value
Proportion of sufficient samples (95% CI)	84% (74–96%)	87% (79–95%)	0.672
Mean ± SD max. diameter of biopsy samples (mm)	1.7 ± 0.81	5.1 ± 2.2	<0.001

CI, confidence interval; E-Bx, endoscopic forceps biopsy; P-Bx, punch biopsy; SD, standard deviation.

**Table 3 diagnostics-11-00360-t003:** Details of unevaluable samples.

	E-Bx (*n* = 7)	P-Bx (*n* = 8)	*p*-Value
Lack of subepithelial interstitium, % (*n*)	100 (7)	100 (8)	n/a
Lack of entire epithelial layer and subepithelial interstitium, % (*n*)	42.9 (3)	0 (0)	0.038 ^1^

^1^ Fisher’s exact test (two-sided). E-Bx, endoscopic forceps biopsy; P-Bx, punch biopsy.

**Table 4 diagnostics-11-00360-t004:** Diagnostic performance of ME-NBI using E-Bx for ≥CIN2.

Sensitivity	92%
Specificity	81%
Positive PV	89%
Negative PV	87%
Accuracy	88%

ME-NBI, magnifying endoscopy with narrow band imaging; CIN, cervical intraepithelial neoplasia; E-Bx, endoscopic forceps biopsy; PV, predictive value.

## Data Availability

Data are not publicly available due to protection of personal data and medical confidentiality.

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
