# Peer review of "Quality Assessment of Endoscopic Forceps Biopsy Samples under Magnifying Narrow Band Imaging for Histological Diagnosis of Cervical Intraepithelial Neoplasia: A Feasibility Study"

_diagnostics, 2021, doi:10.3390/diagnostics11020360_

Round 1
Reviewer 1 Report
The Authors present adaptation of flexible gastrointestinal ME-NBI for diagnosis of CIN (> = 2). They present a feasible diagnostic method that can replace colposcopic cervical biopsy in the future, especially as they are based on many years of experience in studies evaluating the effectiveness of endoscopic biopsies in other organs.The study needs to be validated by direct comparison of methods but the Authors are aware of it.
I have several minor remarks listed below (according to text lines):
66: Are colposcopic findings enough for conization?
65/68: Figure 1 is unclear to me. In the flow of the study presented in this Figure it should be given how many patients were originally screened. The Readers should be informed about a setting of the study? Outpatient? Why 32 patients and not 320, for instance? Were the patients consecutively enrolled? It must be clearly stated in the text that THE SAME PATIENTS underwent colposcopy and ME-NBI YET IN „Study Design, Setting and Participants” chapter.
Figure 1 also mixes entry data and results. Personally I think that these aspects of article should not be located in the same figure.
127: what groups do you mean? I don’t follow… Do you mean, E-Bx and P-Bx groups?
146: There is: „A total of 32 patients underwent ME-NBI examination before conization…” I suggest: „A total of 32 patients underwent ME-NBI examination after colposcopy and before conization…”
Table 1: patients… no „patient”, CIN - cervical intraepithelial neoplasia; In my opinion the word „neoplasm” is limited to CIN3.
What about E-Bx diagnoses?
167: Figure 4a represents forceps… Figure 4b does not represent subepithelial interstitium. (…) It should be „Figure 5 a,b,c” commented in the text.
167-168: You should not repeat data presented in the Table. It is better to describe it in the text.
174: Lower rate of what?
In discussion, the virtual calculation of cost-effectiveness of colposcopy-guided P-Bx vs. ME-NBI guided E-Bx would be interesting.
It is not clear from the content of the article why the sensitivity, specificity, PPV, NPV, accurracy were not performed for the P-Bx group?
Author Response
Response to REVIEWERS' COMMENTS:
Reviewer: 1
Q1. Are colposcopic findings enough for conization?
Response
> Thank you for your prompt query.
As we mention in the ‘Colposcopic Procedure’, colposcopic findings were evaluated according to the Rio 2011 Colposcopy Nomenclature of the International Federation of Cervical Pathology and Colposcopy.
Thus, we believe that colposcopic findings based on the criteria are satisfied for conization.
Q2. 65/68: Figure 1 is unclear to me. In the flow of the study presented in this Figure it should be given how many patients were originally screened. The Readers should be informed about a setting of the study? Outpatient? Why 32 patients and not 320, for instance? Were the patients consecutively enrolled? It must be clearly stated in the text that THE SAME PATIENTS underwent colposcopy and ME-NBI YET IN „Study Design, Setting and Participants” chapter.
Response
> Thank you for your precise suggestion.
As you suggest, the study setting and enrollment seemed obscure for readers. So, we revised in the sessions of Methods and Abstract as follows; ‘We prospectively enrolled 32 consecutive patients who had been scheduled to undergo conization for confirmed or suspected high-grade CIN (≥CIN2) based on colposcopic findings. Next to colposcopy, the same patients underwent ME-NBI just before conization.
Moreover, we revised these unclear parts by dividing Fig.1 into ‘Examination flowchart’ (Fig.1a) and ‘Diagram of study enrollment’ (Fig.1b).
Figure1a. Examination flowchart
Figure 1b. Diagram of study enrollment
Q3. Figure 1 also mixes entry data and results. Personally I think that these aspects of article should not be located in the same figure.
Response
> According to your suggestion, we revised the bothersome Figure 1 as follows; We divided Fig.1 into ‘Examination flowchart’ (Fig.1a) and ‘Diagram of study enrollment’ (Fig.1b).
Furthermore, we revised the related sentence as follows; ‘The examination flowchart and diagram of study enrollment are shown in Figure 1a and 1b, respectively.’
Q4. 127: what groups do you mean? I don’t follow… Do you mean, E-Bx and P-Bx groups?
Response
The two groups indicate E-Bx and P-Bx groups. So we changed the following sentence ‘difference in maximum diameter of biopsy samples between the two groups’ to ‘difference in maximum diameter of biopsy samples between E-Bx and P-Bx groups.’
Q5. 146: There is: „A total of 32 patients underwent ME-NBI examination before conization…” I suggest: „A total of 32 patients underwent ME-NBI examination after colposcopy and before conization…
Response
Thank you for your advice. As you mention, we rewrote as follows; A total of 32 patients underwent ME-NBI examination after colposcopy and before conization in line 147.
Q6. Table 1: patients… no „patient”, CIN - cervical intraepithelial neoplasia; In my opinion the word „neoplasm” is limited to CIN3.
Response
Thank you for your precise comments. We changed the word ‘neoplasm’ to ‘neoplasia’ in Table 1.
Q7. What about E-Bx diagnoses?
Response
Thank you for your question.
As additional analysis, the diagnostic performance of ME-NBI using E-Bx is calculated.
Thus, we changed ‘E-Bx’ to ‘ME-NBI using E-Bx’, and also ‘P-Bx’ to ‘Colposcopy using P-Bx in the related sentences and Table 4. We also revised the whole text.
E-Bx samples under ME-NBI were acquired from lesions suspicious for ≥CIN 2 and non-suspicious lesions. Thus, sensitivity, specificity, PPV, NPV, and accuracy were calculated for the diagnostic ability of ME-NBI. Whereas, P-Bx samples under colposcopy were acquired from only lesions suspicious for ≥CIN 2. Thus, PPV alone was calculated for the diagnostic ability of colposcopy.
Q10. 167: Figure 4a represents forceps… Figure 4b does not represent subepithelial interstitium. (…) It should be „Figure 5 a,b,c” commented in the text.
Response
We made a careless mistake. Thank you for your identification. We revised these figure numbers correctly.
Q11. 167-168: You should not repeat data presented in the Table. It is better to describe it in the text.
Response
Thank you for your advice. We changed the following sentence “The proportion of sufficient samples was 84% (95% CI: 74%–96%) for E-Bx (n=45) and 87% (95% CI: 79%–96%) for P-Bx (n=65), showing no significant difference between the two groups (p=0.672) (Table 2).” to “The proportion of sufficient samples showed no significant difference between E-Bx and P-Bx groups (p=0.672) (Table 2).”.
Q11. 174: Lower rate of what?
Response
According to your suggestion, we added the following sentence ‘・・lower rate of samples without both the entire mucosal layer and subepithelial interstitium’.
Q12. In discussion, the virtual calculation of cost-effectiveness of colposcopy-guided P-Bx vs. ME-NBI guided E-Bx would be interesting.
Response
Thank you for your precise suggestion.
We added the following sentences in the discussion; The instrumental cost of E-Bx under ME-NBI and P-Bx under colposcopy is estimated to be almost equal. Whereas, E-Bx under ME-NBI which may show high diagnostic performance has a potential benefit to reduce the number of biopsy times, leading to its cost-effectiveness over colposcopy-guided P-Bx undergoing random biopsy. Furthermore, patient’s pain accompanied by biopsy may be lower in E-Bx under ME-NBI. Disposal E-Bx forceps would be also advantageous in the view of infection control. A further study is on-going to clarify these potentials.
Q13. It is not clear from the content of the article why the sensitivity, specificity, PPV, NPV, accuracy were not performed for the P-Bx group?
Reponse
Thank you for your advice.
We also answered for query 7.
E-Bx samples under ME-NBI were acquired from lesions suspicious for ≥CIN 2 and non-suspicious lesions. Thus, sensitivity, specificity, PPV, NPV, and accuracy were calculated for the diagnostic ability of ME-NBI. Whereas, P-Bx samples under colposcopy were acquired from only lesions suspicious for ≥CIN 2. Thus, PPV alone was statistically calculated for the diagnostic ability of Colposcopy.
Thus, we added the following sentences in the limitation; P-Bx samples under colposcopy were acquired from only lesions suspicious for ≥ CIN 2, therefore PPV alone was statistically calculated for the diagnostic ability of Colposcopy. Thus, the detailed comparison of ME-NBI vs. Colposcopy was impossible.

Reviewer 2 Report
In this manuscript, the authors aimed to evaluate the feasibility of E-Bx in the diagnosis of CIN. They conducted a clinical study in standard of care cases by comparing it to P-Bx. THe manuscript is well written, and the finding has some significance in clinical practice when the results are further validated in a large sample size. I would recommend to accept it after some minor revision and clarification as follows:
- In histological diagnosis with pathological reading by a pathologist, was the pathologist blinded to the method that was used for biopsy?
- Line 142, the authors mentioned a student's t test for PPV comparison between E-Bx and P-Bx. I do not understand how the authors did, given that PPV is calculated from binary data.
- In table 1, please clarify the final diagnosis. Is it based on pathological examination?
- Given that E-Bx was used to examine the patients who had P-Bx, why were there only 45 patients for E-Bx vs. 65 pts P-Bx? Could the authors explain the reason why another 20 were not tested using E-Bx?
- Table 2, the redundant item of "proportion of sufficient sample" exists.
Author Response
Reviewer 2
Q1. In histological diagnosis with pathological reading by a pathologist, was the pathologist blinded to the method that was used for biopsy?
Response
Thank you for your comments. The pathologist was informed about whether the tissue sample was E-Bx or P-Bx.
So, we added the above sentence in the session of ‘2.4. Evaluation of Biopsy Specimens’.
Q2 Line 142, the authors mentioned a student's t test for PPV comparison between E-Bx and P-Bx. I do not understand how the authors did, given that PPV is calculated from binary data.
Response
Thank you for marking our careless mistake. We used a chi-square test (not a student's t test) for comparing PPV between E-Bx and P-Bx.
Q3. In table 1, please clarify the final diagnosis. Is it based on pathological examination?
Response
Thank you for your precise comments. The final diagnosis was pathological diagnosis of surgical specimens acquired by conization. Then we revised as follows; ‘Final pathological diagnosis of surgical specimens’ in table 1.
Q4. Given that E-Bx was used to examine the patients who had P-Bx, why were there only 45 patients for E-Bx vs. 65 pts P-Bx? Could the authors explain the reason why another 20 were not tested using E-Bx?
Response
We apologize that our obscure sentences probably confuse your understanding.
The same 32 patients underwent both P-Bx under colposcopy and E-Bx under ME-NBI. Thus, we added the following sentences in the sessions of Abstract and 2.1. Study Design, Setting and Participants; ‘Next to colposcopy, the same patients underwent ME-NBI just before conization.’
The numbers of E-Bx and P-Bx biopsy samples for the histological analysis are mentioned in the session of Results; ‘A total of 32 patients underwent ME-NBI examination・・・65 samples were included in the analysis (Figure 1b).’
Thus, we added the Diagram of study enrollment as Figure 1b so that readers can understand easily.
Q5. Table 2, the redundant item of "proportion of sufficient sample" exists.
Response
Thank you for marking our careless mistake. Thus, we deleted the redundant item of "proportion of sufficient sample" in table 2.

Round 2
Reviewer 1 Report
Thank you for corrections.
My minor remarks:
- Please remove "just before conization" from Figure 1a (second frame).
- Line 242: please remove "random biopsy" from this passage.
Congratulations of the article.
Author Response
Response to REVIEWERS' COMMENTS:
Reviewer: 1
Q1. Please remove "just before conization" from Figure 1a (second frame).
Response
> According to your suggestion, we revised the Fig.1a.
Q2. Line 242: please remove "random biopsy" from this passage.
Response
> According to your suggestion, we removed "random biopsy".
We appreciate for spending your precise time and your helpful suggestion.
Reviewer 2 Report
No more comments are made
Author Response
Reviewer: 2
Thank you very much for spending your precise time and your helpful suggestion.
